# Understanding Development of Malnutrition in Hemodialysis Patients: A Narrative Review

**DOI:** 10.3390/nu12103147

**Published:** 2020-10-15

**Authors:** Sharmela Sahathevan, Ban-Hock Khor, Hi-Ming Ng, Abdul Halim Abdul Gafor, Zulfitri Azuan Mat Daud, Denise Mafra, Tilakavati Karupaiah

**Affiliations:** 1Dietetics Program, Faculty of Health Sciences, Universiti Kebangsaan Malaysia, Jalan Raja Muda Abdul Aziz, Kuala Lumpur 50300, Malaysia; sham_0901@yahoo.com; 2Department of Medicine, Faculty of Medicine, Universiti Kebangsaan Malaysia, Jalan Yaakob Latif, Bandar Tun Razak, Cheras, Kuala Lumpur 56000, Malaysia; khorbanhock@gmail.com (B.-H.K.); halimgafor@gmail.com (A.H.A.G.); 3School of Medicine, Faculty of Health & Medical Sciences, Taylor’s University Lakeside Campus, No 1, Jalan Taylors, Subang Jaya 47500, Malaysia; nghiming@gmail.com; 4Department of Dietetics, Faculty of Medicine & Health Sciences, Universiti Putra Malaysia, UPM Serdang 43400, Malaysia; zulfitri@upm.edu.my; 5Post Graduation Program in Medical Sciences and Post-Graduation Program in Cardiovascular Sciences, (UFF), Federal Fluminense University Niterói-Rio de Janeiro (RJ), Niterói-RJ 24033-900, Brazil; dmafra30@gmail.com; 6School of BioSciences, Faculty of Health & Medical Sciences, Taylor’s University Lakeside Campus, No 1, Jalan Taylors, Subang Jaya 47500, Malaysia

**Keywords:** malnutrition, hemodialysis, iatrogenic, non-iatrogenic factors

## Abstract

Hemodialysis (HD) majorly represents the global treatment option for patients with chronic kidney disease stage 5, and, despite advances in dialysis technology, these patients face a high risk of morbidity and mortality from malnutrition. We aimed to provide a novel view that malnutrition susceptibility in the global HD community is either or both of iatrogenic and of non-iatrogenic origins. This categorization of malnutrition origin clearly describes the role of each factor in contributing to malnutrition. Low dialysis adequacy resulting in uremia and metabolic acidosis and dialysis membranes and techniques, which incur greater amino-acid losses, are identified modifiable iatrogenic factors of malnutrition. Dietary inadequacy as per suboptimal energy and protein intakes due to poor appetite status, low diet quality, high diet monotony index, and/or psychosocial and financial barriers are modifiable non-iatrogenic factors implicated in malnutrition in these patients. These factors should be included in a comprehensive nutritional assessment for malnutrition risk. Leveraging the point of origin of malnutrition in dialysis patients is crucial for healthcare practitioners to enable personalized patient care, as well as determine country-specific malnutrition treatment strategies.

## 1. Introduction

The last three decades witnessed considerable growth in the global burden of chronic kidney disease (CKD), accounted for by 77.5% of end-stage kidney disease (ESKD) patients on kidney replacement therapy (KRT), with 43.1% alone provided by dialysis [1]. Hemodialysis (HD) forms 89% of the global treatment for ESKD patients [2]. The technological delivery of HD treatment to patients today is considered optimal as per medical guidelines for practice with regard to biocompatibility of dialyzer membranes, dialysis dose, frequency of dialyzer reuse, and duration of dialysis [3]. A significant problem faced by this patient group, however, is malnutrition with a global prevalence of 28–54% [4], facing a greater risk of mortality, varying from a likelihood of 1.61 to 4.08 [5,6].

Morbidity arising from malnutrition in these patients severely affects quality of life (QoL) [7], frailty, and increased risk of infections and mortality [8]. Malnutrition in patients on dialysis develops along different pathways from that observed in acute cases of hospitalization and critical illness. Its inception evolves from the early progressive nature of CKD itself [9], the implementation of a low protein diet to limit CKD progress [10], and the prolonged period of potentially lifesaving dialysis treatment for patients reaching ESKD [11]. In this context, the HD treatment itself in terms of dialysis-induced nutrient losses, multiple dialyzer reuse, dialysis-induced inflammation, efficacy of uremia and metabolic acidosis correction, and dialysis adequacy, frequency, and duration are inevitable iatrogenic factors contributing to malnutrition. Concurrently, prevailing non-iatrogenic factors such as suboptimal dietary intakes, taste alterations, poor appetite, insulin resistance, and psychosocial factors are also incriminated in the etiology of malnutrition.

Malnutrition occurrence in the dialysis population has generated much research. However, different definition terms exist for malnutrition such as protein-energy wasting, protein-energy malnutrition, malnutrition–inflammation complex syndrome, malnutrition–inflammation–atherosclerosis and uremic wasting syndrome depending on involvement of inflammation, hypercatabolism, and increased uremia [12]. Multiple factors are cited within the etiology of these descriptive malnutrition terms, and implication of some but not all these factors differentially indicates that there is no uniformity in the diagnosis of malnutrition [12]. Comorbidities such as heart failure (left-ventricular failure) and CKD-mineral bone disorder have a bidirectional association with nutritional status [13,14]. The related mechanisms include malabsorption due to gut edema, poor appetite due to cytokine production, and difficulty in oral intake and food preparation arising from fatigue and breathing difficulty [15]. However, malnutrition is acknowledged to be a kidney-specific risk factor for heart failure in CKD patients [14], and, with HD patients, there is lack of association between malnutrition assessment and echocardiographic findings [16] or cardiovascular disease (CVD) risk [17].

In essence, factors contributing to the development of malnutrition may be categorized as of iatrogenic and non-iatrogenic origin (Figure 1). Iatrogenic factors are an inadvertent consequence of dialysis for ESKD patients, whereas non-iatrogenic factors develop spontaneously from contributive factors accompanying the progression of CKD but not related to the primary treatment.

This review aims to provide a global view of the iatrogenic and non-iatrogenic causes of malnutrition, irrespective of its form described in current literature, as some aspects with regard to this topic have changed over recent years. Our approach discusses mechanisms elucidating the implication of each factor contributing to malnutrition. The origin of malnutrition contributed by both iatrogenic and non-iatrogenic factors is important to understand the implications for healthcare practitioners in performing the assessment and treatment for malnutrition. Before this categorization, malnutrition must also be delineated according to its development history. This review is, therefore, organized into (i) development of malnutrition in HD, (ii) iatrogenic factors of malnutrition, and (iii) non-iatrogenic factors of malnutrition.

## 2. Development of Malnutrition at the Time of HD Initiation and Indicators of Poor Nutritional Status

The decision to start dialysis for an ESKD patient varies across countries and is influenced by the local nephrology practice, healthcare policies, and cost for dialysis treatment [18]. The Dialysis Outcomes and Practice Patterns Study (DOPPS) Phase 2 with 12 participating countries indicated a greater mortality rate in patients new to dialysis compared to prevalent dialysis patients [19]. Early mortality at the time of dialysis initiation prevails with increased risk up to 80% within the first two months of HD initiation [20]. Apart from catheter vascular access [20] and pre-dialysis care [21], nutritional status is considered a potentially modifiable risk factor in early mortality [22]. Clearly, pre-existing malnutrition originates from progressive CKD stages 3 to 5 with vulnerability of the patient starting from the point of metabolic derangements associated with falling glomerular filtration rate, late nephrology access, and insufficient pre-dialysis dietetic care during this period [22,23].

Earlier opinion on dialysis initiation did consider poor nutritional status as a factor to initiate dialysis. However, this was not based on markers of malnutrition but rather signs and symptoms of malnutrition such as anorexia, nausea, and fatigue [24]. Van de Luijtgaarden et al. (2012) reported 53% of nephrologists agreeing to initiate dialysis in patients with poor nutritional status [25]. However, a review on dialysis initiation observed a lack of data on the benefits of early dialysis initiation in patients with low serum albumin level or in improving nutritional status [26]. The Canadian Society of Nephrology Guidelines (2014) [27] ceased to recommend dialysis initiation on the basis of a decline in nutritional status as indicated by serum albumin, lean body mass, or subjective global assessment (SGA), whereas the Caring for Australians with Renal Impairment Guidelines [28] recommend dialysis with glomerular filtration rate (GFR) < 10 mL/min per 1.73 m^2^ to reduce uremic symptoms or signs of malnutrition.

Dialysis treatment is expected to improve nutritional status for patients with a more liberal protein prescription compared to the pre-dialysis stage [29,30,31]. However, dialysis treatment is cited to contribute to malnutrition burden [8], and newly dialyzing patients are at risk of the early mortality attributed to malnutrition as evidenced by diagnostic assessment of nutrition risk screening using SGA [32], low body mass index (BMI), low mid-arm muscle circumference (MAMC) [22,33,34], low albumin [20], low cholesterol levels [32], and reduced food intake [22,33,34,35], as shown in Table 1.

## 3. Iatrogenic Factors of Malnutrition

ESKD patients with pre-existing malnutrition on maintenance dialysis become additionally vulnerable over time to the catabolic effects of the dialysis treatment, which predispose the patient to greater mortality and morbidity in long-term dialysis. The concern is that the presence of poor nutritional status in dialysis patients predicts increased mortality risk.

Kwon et al. (2016) prospectively used SGA to monitor patients, observing that those in SGA B and C categories at baseline almost tripled their risk of mortality by 12 months [32]. HD patients experiencing declines in BMI and serum albumin levels over a 6 month follow-up also had increased mortality risk [36]. Table 2 summarizes studies reporting various indicators of poor nutritional status as strong predictors of mortality in maintenance HD patients.

Iatrogenic malnutrition or “physician-induced malnutrition” is the development of malnutrition arising from medical procedures, pharmacological treatment, prolonged hospitalization, nosocomial infections, or delayed wound healing [40]. Similarly, aspects of the dialysis procedure contribute to malnutrition, which is unavoidable as it occurs as part of the treatment [8]. The iatrogenic aspects of dialysis procedure are detailed in the sections below.

### 3.1. Dialysis-Induced Nutrient Losses

The dialysis process is instrumental to chronic nutrient losses, particularly protein and amino acids. About 6–12 g of amino acids and 7–8 g of protein losses occurring during each dialysis session [41,42,43,44,45] may contribute to hypoalbuminemia, a strong predictor of malnutrition and mortality [11,33,36]. Optimal dietary protein intake (DPI) may replenish low plasma amino acids. However, DPI inadequacy is a common issue in HD patients, affecting 32–81% of HD populations globally [31]. Suboptimal DPI associated with dialysis-induced amino-acid losses [46] promotes protein catabolism through increased whole-body and muscle protein proteolysis [46,47].

Nutrient losses via dialysis depend on the mechanism of solute removal and the pore size of the dialyzer membrane, which determines solute removal [48]. However, increasing the pore size of dialyzer membranes to enable greater removal of middle molecules also increases involuntary albumin losses, estimated between 2 and 14 g depending on the degree of membrane permeability [49]. As such bioincompatible membranes [45], high flux membrane, hemofiltration (HF) and hemodiafiltration (HDF) techniques [46,50], or multiple dialyzer reuse practice [43] induce greater membrane permeability and facilitate greater losses of amino acids into the dialysate [50]. Table 3 shows the degree of protein and amino-acid losses associated with membranes characteristics.

Dialysis performed during the 1960s used low-flux membranes [55], which efficiently removed uremic solutes with low molecular weight (<0.5 kDa) but not the middle molecules of 0.5–60 kDa size [56,57]; with advanced technology, greater removal of larger uremic solutes using high-flux membranes and, now, membranes with medium (MCO) and high cutoffs (HCO) are available [53], combining larger pore sizes with improved HF and HDF techniques [41,58].

With highly permeable membranes or HF and HDF techniques, patients are reported to achieve better intradialytic and hemodynamic stability [59], alongside improvements in nutritional status as evidenced by gains in BMI, dry weight, and appetite [60]. Improvements are attributed to greater removal of middle molecules using HDF. However, these membranes and/or techniques, in addition to increasing the cost burden [57,59,60,61,62], induce greater albumin losses of 3.5 to 9.0 g per HD session [63,64], along with involuntary removal of vitamins, larger protein molecules, and lipids [62].

Contrarily, two studies reported a significant reduction in serum albumin levels in patients dialyzing with HCO membranes [63,65]. The risk–benefit balance by using dialyzer membranes with greater permeability for improved uremic solute removal versus greater albumin losses remains unknown [41]. Similarly, advanced techniques using either HF or HDF may also pose long-term risk of malnutrition development in HD patients. Given that the permissible threshold for tolerance of albumin losses with highly permeable membrane remains unclear, long-term use of MCO [66] or HCO membranes with HDF [63] techniques may pose malnutrition risk.

### 3.2. Multiple Dialyzer Reuse

In low-to-middle-income countries, the practice of dialyzer reuse is common [67,68]. However, multiple dialyzer reuse may contribute to negative outcomes [69] such as infection risks, biochemical and immunologic reactions, improper sterilization, increased membrane permeability [49], and loss of performance leading to inadequate dialysis adequacy. These issues are believed to arise from the reprocessing procedure involving sanitizing agents [67,70]. However, two studies have indicated that single, minimal (>6 times), or multiple dialyzer reuse carries no impact on dialysis adequacy, body weight, and serum albumin level [71,72].

### 3.3. Dialysis-Induced Inflammation

Many factors lead to inflammation in HD patients such as biocompatibility of the dialyzer membrane [73,74,75], infection related to dialysis access [73], and impure dialysate containing cytokine-inducing substances labeled as endotoxins [73,76].

Whether dialysis access directly contributes to malnutrition has not been shown. Arteriovenous fistula (AVF) failures were not influenced by markers of poor nutritional status except for high cholesterol (*p* = 0.034) and low normalized protein catabolic rate (*n*PCR) levels (*p* = 0.029) [77]. HD patients with catheter access compared to fistula and graft had significantly higher malnutrition–inflammation score (MIS) and lower serum albumin levels [78]. In fact, patients with AVF have 52% greater survival rate compared to those on central venous catheter (CVC) irrespective of nutritional status, although malnutrition was found to lower survival rate by 2% [79]. Instead, catheter rather than graft and fistula access appears to be a significant predictor of greater inflammatory response, and it is associated with the highest all-cause mortality rate [78] mediated by infection [20]. Therefore, the route of dialysis access is rather associated with inflammation and mortality risk [78], whereby presence of malnutrition may influence the survival rate [79].

Direct effects of the membrane, the extent of complement stimulation induced by the membrane, and the degree of eosinophilia associated with the clearance of cytokines determine the magnitude of the inflammatory response during dialysis [74]. Inflammatory marker levels may be modulated by different types of dialyzer membranes (Table 4). Generally, the high-flux dialyzer membrane and HDF technique are associated with lower inflammation grade in HD patients when compared to the low-flux dialyzer membrane. These differences are attributed to processing technology for structuring and composition of the membrane, conferring attributes to the dialyzer in terms of biocompatibility, water permeability, clearance, and appropriate sieving coefficients for myoglobin or albumin [59].

Inflammation also occurs with dialysate contamination by microorganisms, which produce endotoxins that pass through the dialyzer membrane and enter into blood circulation [83], amplifying the production of proinflammatory cytokines [84] such as interleukin (IL)-1, IL-6 [85], and tumor necrosis factor (TNF)-α [86]. Infected or old clotted grafts may also contribute to inflammation [87]. Of note, these middle molecules such as IL and TNF-α are not effectively removed by dialysis treatment with low-flux membrane [88] and are accumulated.

Overall, dialysis patients are vulnerable to oxidative stress with a marked increase in reactive oxygen species (ROS) production and antioxidant depletion. ROS induces activation of nuclear factor kappa B (NF-κB), which is translocated to the cell nucleus stimulating cytokine production, in turn causing inflammation [89]. Indeed, HD patients have pronounced NF-κB gene expression compared to a healthy population [90].

Another impact of the HD treatment is the activation of polymorphonuclear white blood cells, which trigger production of ROS and other pro-oxidants [91]. Indeed, increased indices of oxidative damage along with decreased indices of antioxidant defense have been observed in HD patients post-dialysis [92]. Low antioxidant levels in HD patients may also occur from limited vegetable and fruit intakes preventing hyperkalemia [93]. Resultant low intakes of vitamin A, C, and E and selenium would affect antioxidant defense mechanisms [93,94]. Additionally, involuntary removal of vitamins also occurs with every HD session as mentioned in Section 3.1.

However, Silva et al. (2019) found no difference in markers of oxidative stress and antioxidant defenses in malnourished HD patients identified using global objective assessment compared to mild or well-nourished patients [95]. In contrast, accumulation of advanced glycation end products, a biomarker of oxidative stress measured using skin auto fluorescence, was significantly associated with markers of malnutrition in HD patients such as lower serum albumin, lower handgrip strength, and lower protein intake [96]. HD patients with protein-energy wasting were 5.2 times more likely to experience oxidative stress as demonstrated by high protein carbonyl levels (95% confidence interval (CI): 24.0–1.1, *p* = 0.039) [97].

When malnutrition coexists with inflammation in dialysis patients, the combination of both conditions is known as malnutrition–inflammation complex syndrome [98]. The inflammation results in a reduction in albumin production in the liver [99] and fosters poor appetite, a non-iatrogenic factor implicated in malnutrition [100]. The strong relationship between malnutrition and inflammation in dialysis patients is evident, as indicated by significant association (*r* = 0.65, *p* = 0.040) of malnourished HD patients categorized by SGA B and C with high C-reactive protein (CRP) levels [101].

### 3.4. Efficacy of Uremia Correction

A major aim of dialysis therapy is to remove uremic waste products [57,102]. However, dialysis only reduces uremic burden through partial removal of uremic solutes [103]. Incomplete clearance of uremic solutes and the generation of urea from both dialysis-induced tissue degradation and dietary proteins contribute to uremic solute accumulation [104]. Excessive uremic solute burden influences amino-acid and protein metabolism by inhibiting transamination activities of enzymes such as threonine dehydratase and alanine and aspartate transferases [104], impairing membrane transport [105], inhibiting protein binding [106], and promoting muscle wasting [104].

The removal of uremic solutes depends on dialyzer membrane permeability. As mentioned earlier (see Section 3.1), uremic solutes with low molecular weight such as urea and creatinine are efficiently dialyzed via a low-flux dialyzer membrane [88], whereas membranes with higher permeability allow for greater clearance of small and large middle molecules. However, the threshold for clearance depends on uremic gains on non-dialysis days and total clearance achieved from the previous HD session [104,107]. Both these factors would determine the severity of uremia in HD patients. Aside from the permeability of the membrane, total clearance of uremic solutes is also determined by dialysis adequacy, dialysis frequency, and duration of dialysis session [108].

### 3.5. Dialysis Adequacy

Uremic solute clearance depends on dialysis adequacy, which refers to the frequency and duration of dialysis [102]. Expert guidelines for optimal uremic solute removal favor a three- to five-hourly dialysis session provided three times weekly [3] in order to meet dialysis adequacy by achieving a Kt/V_urea_ of 1.2, which designates the dialyzer urea clearance (K), time on dialysis (*t*), and total body water (V) [57,102].

However, the hemodialysis (HEMO) study in a 3 year follow-up of 1846 dialyzing patients established that neither high (Kt/V = 1.65) nor low (Kt/V = 1.25) dialysis dose significantly affected markers of nutritional status such as serum albumin, post-dialysis weight, dietary energy and protein intakes, calf and upper arm circumference, and appetite status (all *p* > 0.05) [109]. The only effect was on normalized protein catabolic rate (*n*PCR), a surrogate marker for DPI with greater decline in the low compared to high dialysis dose group (*p* = 0.007).

Of note, the calculation of Kt/V_urea_ is based on urea, a surrogate marker for clearance of small solutes. It does not represent removal of the more detrimental larger uremic solutes [107,110]. Inefficient removal of uremic toxins via dialysis is suggested to induce taste alteration in HD patients, which contributes to malnutrition [111,112].

### 3.6. Dialysis Frequency

Increasing the thrice-weekly frequency of dialysis sessions to >4 times may support better management of fluid removal [113] and lower systolic blood pressure, as well as improve QoL [23]. Alternately, a shorter but increased frequency of dialysis provided by six HD sessions two-hourly per week may benefit toward greater removal of uremic solutes [56], as shown in patients with lower trends in pre-dialysis serum levels of creatinine, urea, uric acid, and protein-bound solutes such as indole-3-acetic acid and indoxyl sulfate. However, serum albumin levels and post-dialysis weight did not improve for these patients [56]. In contrast, Rashidi et al. (2011) observed improved weight, BMI, and serum albumin status along with decreased serum CRP in patients converting to four HD sessions four-hourly per week from the standard dialysis regime by six weeks [114]. Dietary intake also improved albeit non-significantly. This effect may perhaps be explained by improved appetite occurring with greater removal of uremic compounds through more frequent dialysis [115].

In some countries, weekly frequency of dialysis may depend on the patient’s access to financial support. For example, dialysis frequency in low-income countries such as India and Pakistan may be offered as two sessions four-hourly per week [116,117] compared to the standard dialysis of three sessions four-hourly per week in developing countries [116,118]. Treatment affordability, poor access to nephrology care and dialysis centers [116], and inadequately equipped dialysis facilities [117] are reasons for lower dialysis frequency. Interestingly, Chauhan and Mendonca (2015) showed that, for 50 Indian HD patients undergoing dialysis twice a week, those achieving dialysis adequacy of Kt/V ≥ 2.0 significantly improved their serum albumin and hemoglobin levels [118].

### 3.7. Dialysis Duration

Increasing the dialysis duration results in greater removal of small and large uremic solutes compared to the standard HD regime [102,119]. Patients dialyzing for 8 h using a high-flux dialyzer membrane have shown greater total solute removal, dialyzer extraction ratios, and total cleared volumes for urea, creatinine, phosphorus, and β_2_-microglobulin compared to patients on standard dialysis, and this occurs without affecting dialysis adequacy [102]. Lower post-dialysis levels of the uremic toxin indoxyl sulfate have been observed in patients dialyzing for 8 h compared to the standard 4 h regime (17.2 ± 3.6 vs. 27.5 ± 3.2 g/mL, *p* = 0.049), despite both patient groups having similar pre-dialysis levels [120].

Nutritional marker improvements through higher serum albumin and hemoglobin levels and lower white blood cell count appear to be associated with longer dialysis duration, as indicated from combined data of the three DOPPS [121]. These improvements may be explained by greater removal of both small and large solutes with longer hours of dialysis [102].

### 3.8. Efficacy of Metabolic Acidosis Correction

Metabolic acidosis develops in the early stages of CKD from the kidney’s inability to excrete nonvolatile acids and synthesize bicarbonate to maintain acid–base balance [122]. HD treatment aims to correct metabolic acidosis via bicarbonate concentration of the dialysate [123], ultrafiltration rate [124], dialyzer membrane surface area and permeability [125], blood and dialysis flow rate [124], transmembrane concentration gradient set by the patient’s serum bicarbonate level and bicarbonate availability from the dialysate [125], and dialysis adequacy [122,123] through maintaining the pre-dialysis serum bicarbonate levels between 24 and 26 mmol/L as recommended by current opinion [126]. However, metabolic acidosis correction depends on patient-related determinants such as interdialytic weight gain [123], acid generation from high protein intake [122,127], or gastrointestinal losses of bicarbonate [122,125]. Individual fluctuation in patients’ bicarbonate levels challenges optimum management [122].

Metabolic acidosis contributes to malnutrition by reducing protein synthesis and increasing muscle degradation [123]. The malnutrition pathway in HD patients involves protein catabolism, secondary insulin resistance, inflammation, and increased serum leptin levels [122]. Lines of evidence using animal and human studies explain that increased muscle breakdown occurs during metabolic acidosis via two mechanisms. These involve increased activation of branched-chain ketoacid dehydrogenase (BCKAD) and the ATP-dependent ubiquitin–proteasome system (UPS) pathway [128]. Importantly, acidosis stimulates increased gene transcription and activity of BCKAD enzyme to degrade the branched-chain amino acids (BCAA), namely, leucine, isoleucine, and valine. BCAAs are important precursors for protein synthesis and are mainly metabolized in the muscle [50]. Increased BCAA oxidation, therefore, is the basis for a higher protein requirement for HD patients [129]. However, metabolic acidosis concomitant with dietary insufficiency and uremia further exacerbates protein catabolism in dialysis patients [130]. Metabolic acidosis activates UPS by increasing gene transcription of the proteasome and ATP-dependent ubiquitin, components involved in the muscle protein degradation pathway [128]. This chain leads to increased caspase-3 activity which promotes cleaving of muscle fibers, resulting in poor muscle mass [128].

Additionally, the acidic environment affects insulin binding to receptors, thus reducing tissue sensitivity to insulin and affecting glucose uptake [131]. Separately, metabolic acidosis also inhibits the anabolic effect of insulin, causing muscle depletion in dialysis patients [122]. Moreover, cell culture studies have shown that TNF-α and interleukins are generated in an acidic environment, triggering an inflammatory response [132,133].

The impact of metabolic acidosis on nutritional status of HD patients by assessment of serum bicarbonate levels may present anomalies in interpretation. In one study, patients with serum bicarbonate levels ≤22 mmol/L rather than serum bicarbonate levels >22 mmol/L had lower serum albumin levels (*p* = 0.046) [134]. In these patients, high serum bicarbonate levels correlated negatively with *n*PCR (*r* = −0.492, *p* = 0.045) but positively with serum albumin (*r* = 0.432, *p* = 0.019). Acidosis-led catabolism triggers breakdown of the endogenous proteins, which influence higher *n*PCR levels. In different malnourished HD populations, serum bicarbonate levels of >23 [135,136] or >27 mmol/L [127,136] have been associated with greater mortality risk. As malnutrition is a confounding factor for serum bicarbonate level, there is no ideal serum bicarbonate level that fits all dialysis patients [137].

## 4. Non-Iatrogenic Causes of Malnutrition

Comorbid non-iatrogenic factors may also contribute to malnutrition development in dialysis patients. These non-iatrogenic factors are elaborated on in the sections below.

### 4.1. Suboptimal Dietary Intake

Suboptimal dietary intake is a primary contributing factor to malnutrition [12] and is associated with increased mortality in HD patients [31,138]. Adult recommendations for dietary energy intake (DEI) and DPI to achieve nutrient adequacy have been proposed for HD patients by several expert groups, and these generally fall within 25–35 kcal/kg ideal body weight (IBW)/day for DEI and 1.0–1.2 g protein/kg IBW/day for DPI [126,139,140,141]. Requirements factor in criteria to maintain physiological balance, prevent deficiencies from dialysis-induced nutrient losses, and reduce risk of malnutrition and mortality [126].

However, achieving DEI and DPI adequacies remains a challenge for HD patients with intakes falling below recommendations as indicated by many studies (Table 5). This is evidenced by 70–90% of global HD populations reported with DEI inadequacy, whereas DPI inadequacy ranges between 30% and 80%.

Suboptimal DEI is of greater concern than DPI inadequacy, as gluconeogenesis is implied. Three studies reported HD patients achieving DPI adequacy >1.2g/kg/BW but failing to meet DEI adequacy [146,151,155]. Insufficient DEI, despite DPI adequacy, predisposes patients to negative nitrogen balance, resulting in both dietary protein and muscle protein to be diverted to fuel body energy requirements [154]. Additionally, amino-acid losses occurring through the dialysis procedure (see Section 3.1) affect protein synthesis, triggering muscle proteolysis to generate amino acids if there is low DPI [158]. Of concern, suboptimal dietary intake bears a negative impact on the survival rate of HD patients as indicated by some studies reporting patients with poor DEI and DPI (Table 6). Kang et al. (2017) found that HD patients with DEI < 25 kcal/kg BW/day and DPI < 0.8g/kg BW/day had 86% and 35% increased risk of mortality, respectively [138]. Similarly, those with DPI < 1.2g/kg BW/day had a 4.98-fold greater risk of mortality [159]. A recent metabolomics study reported higher concentrations of 3-hydroxybutyrate and tartrate along with low creatinine appearing in patients with protein energy wasting [160]. These metabolites are linked to gluconeogenesis and may be conditional to suboptimal DEI and DPI intakes [161].

The background of dietary inadequacy observed in HD patients may be attributed to monotonous dietary patterns [142,154,162], poor diet quality [154], anorexia [154], and alterations in taste [163]. A monotonous diet defines a dietary pattern with minimal variety of food groups [142,154]. Zimmerer et al. (2003) showed that HD patients with the highest Diet Monotony Index (DMI) had the lowest DEI (21 kcal/kg/day) and DPI (0.83 g/kg/day) [162]. Furthermore, a 5 point increase in DMI was associated with decreases in both energy and protein intakes by 10 kcal/kg BW/day (*p =* 0.004) and 0.43 g/kg BW/day (*p* = 0.006), respectively. Importantly, Sualaheen et al. (2020) investigating habitual dietary patterns of Malaysian HD patients showed that the highest vs. lowest tertiles of a balanced and varied dietary pattern were associated with lower DMI (T3 = 29.0 ± 1.1 vs. T1 = 33.0 ± 1.0, *p =* 0.030) and lower malnutrition risk identified via malnutrition–inflammation score (MIS) assessment (T3 = 4.9 ± 0.36 vs. T1 = 6.4 ± 0.34, *p =* 0.010) [142].

Suboptimal DEI from reduced food intake also affects patient adequacy for other essential nutrients, as the overall diet quality falls. Kim et al. (2015) reported that an insufficient DEI of 21.90 ± 6.70 kcal/kg BW affected adequacy for micronutrients such as vitamin A and C, thiamin, riboflavin, niacin, folate, calcium, phosphorus, and zinc, as well as dietary fiber [154].

### 4.2. Taste Alterations

Low palatability of diets is underscored by taste alterations experienced by HD patients, and this factor reportedly affects 31–44% of HD populations [165]. Lynch et al. (2013) found 34.6% of 1745 HD patients in the HEMO study self-reporting taste alteration [165]. Such patients compared to those reporting “no taste alterations” clearly signified poor nutritional status as indicated by lower dry weight, serum albumin, serum creatinine, *n*PCR, and DPI. Reported DEI values for both groups were similarly low (22.8 ± 9.8 vs. 23.1 ± 9.4 kcal/kg/day, *p* = 0.260) highlighting that energy inadequacy evidently prevailed for all patients. This study also reported a 71% increased risk of mortality (odds ratio (OR) = 1.71, 95% CI: 1.01–1.37) in patients with altered taste perception.

Taste alterations experienced by HD patients may be explained by food aversion learning [166]. Aversions toward protein-rich foods such as meat have been significantly associated with enhanced metallic taste in patients also reporting poor appetite [167]. Clearly, taste alterations in HD patients develop food aversion learning, which impacts appetite and reduces overall diet quality, thus contributing to malnutrition. The reduction in taste perception may also be related to zinc deficiency [168].

### 4.3. Poor Appetite

HD patients reporting poor appetite experience significantly higher frequency of hospital admissions, longer duration of hospitalization, poor QoL, and nutritional outcomes such as lower normalized protein nitrogen appearance levels and high inflammatory marker levels than those reporting good appetite [163,165]. The immediate impact of poor appetite is reduced dietary adequacy and increased risk of malnutrition. This was shown in Malaysian HD patients where poor appetite compared to good appetite was significantly linked to lower DEI (14.34 vs. 23.12 kcal/kg/IBW/day, *p* = 0.049), DPI (0.45 vs. 0.94 g/kg IBW/day, *p* = 0.010) but higher MIS scores (9.5 vs. 6.6, *p* = 0.039) indicative of malnutrition [148]. Of concern, patients with diminished appetite faced 4.74 times greater risk of mortality [163].

The mechanism for poor appetite may be explained by changes in appetite hormones in HD patients. Ghrelin, an orexigenic hormone mainly secreted by the stomach, regulates appetite by stimulating spontaneous food intake [29,169]. Ghrelin present in its active form as des-acyl ghrelin has an anorexigenic effect, whereas acyl ghrelin as ghrelin in its inactive form is the main orexigenic molecule [169]. Together, these studies suggest that des-acyl ghrelin may have a negative effect on appetite, whilst high acyl ghrelin levels associated with adiposity indicate better nutritional status [29,169]. Moreover, an association among ghrelin, inflammation, and nutritional status has been reported [170].

Leptin, an adipokine, has an inhibitory effect on appetite in normal metabolism [171]. However, leptin’s role in regulating appetite in CKD is controversial. Hypoleptinemia has been associated with malnutrition in HD populations although its mechanistic involvement in causing poor nutritional status is unknown [172,173]. Low serum leptin levels were independently associated with high MIS status observed in 100 Taiwanese HD patients [172] and 65 Turkish HD patients [174]. However, these studies could not show any association of low leptin levels with inflammatory markers. It may be implied that low leptin levels may serve as a marker of poor nutritional status, whereas higher levels may indicate leptin resistance [175], as there is an attenuation of appetite suppression [172]. Iikuni et al. (2008) proposed leptin as the link between energy homeostasis and inflammation in normal metabolism, where greater leptin secretion occurs with greater adiposity, which in turn promotes production of inflammatory cytokines [176]. Applying this hypothesis to the CKD population, it may be inferred that malnourished HD patients with low BMI will have less leptin secreted by adipocytes, whereas the reverse may occur with better nutritional status and higher BMI.

Studies reporting the impact of poor appetite on malnutrition as indicated by various nutritional outcomes in HD patients are summarized in Table 7.

### 4.4. Insulin Resistance

Insulin resistance is implicated in the etiology of malnutrition in HD patients. Insulin at physiological levels bears both catabolic and anabolic effects on skeletal muscle. Insulin’s anabolic role is to promote BCAA transport and regulate protein synthesis in the muscle [180]. Another anabolic role of insulin is facilitating glucose transport and uptake by muscle tissues [181]. Reduced insulin secretion by the pancreatic β-cells or impaired tissue sensitivity to insulin at receptor and post-receptor levels in the heart, liver, or muscle are two pathways of insulin insufficiency [182,183]. More commonly, “uremic insulin resistance” through inflammatory pathways may occur from insufficient removal of dialyzable uremic solutes [180]. Of relevance, insulin resistance at receptor levels are traced to defects in the insulin receptor signaling pathway arising from metabolic derangements accompanying kidney disease such as uremia, metabolic acidosis, anemia, and inflammation [8,184].

Insulin resistance is associated with peripheral resistance of glucose uptake at the skeletal muscle site and manifests as impaired insulin signaling through the phosphorylation of insulin receptor substrate-1, which inhibits tyrosine kinase activity at the insulin receptor [181,183]. Another pathway of insulin resistance and impaired glucose metabolism may be suggested by high circulating retinol binding protein 4 (RBP4) [185]. Animal studies have demonstrated that RBP4′s inverse relationship with glucose transporter type 4 (GLUT4), which is insulin-dependent, induces glucose uptake in the fat and muscle [186,187]. High RBP4 has a role in glucose metabolism by inducing gluconeogenesis and inhibiting glucose uptake in the muscle via feedback suppression of adipose tissue expression, followed by reduced GLUT4 expression, which affects glucose uptake [185].

Reduced insulin sensitivity affects BCAA transport, blunting the anabolic effect of insulin for decreasing skeletal muscle breakdown [180]. Depletion in BCAA due to amino-acid losses via dialysis, along with suboptimal dietary intake lead to increased proteolysis to supply amino acids needed for protein synthesis [188]. Therefore, insulin resistance promotes muscle proteolysis, and this association is evident in HD patients with studies reporting a positive correlation between insulin resistance and muscle loss [189,190].

It is, thus, clear that, in ESKD patients, apart from chronic suboptimal food intake, gluconeogenesis may also be driven by insulin resistance associated with inflammation, uremia, and metabolic acidosis. Gluconeogenesis is a normal adaptive catabolic process to produce energy. Protein sparing under conditions of energy sufficiency occurs as the primary protein function for tissue synthesis and repair. With dietary energy insufficiency, amino acids and proteins derived from dietary protein or breakdown of skeletal muscle during starvation become new substrates for energy [158]. As HD patients are known to have suboptimal food intake, increased gluconeogenesis in these patients stimulates muscle proteolysis, leading to greater risk of malnutrition.

### 4.5. Psychosocial Factors

Psychosocial factors may negatively impact physical and emotional status, QoL, and nutritional status in HD patients [149,191,192], as shown in Table 8.

#### 4.5.1. Depression

Depression is reported to be prevalent in 6% to 84% of HD patients [204] and arises from loss of the provider role within a family, unemployment, lack of social support, reduced mobility, physical strength, cognitive ability, and sexual function [204]. Additional factors are anxiety and stress from the burden of kidney failure followed by fluid and dietary restrictions, which are significantly associated with poor QoL in these patients [204,205].

Several studies have reported associations between depression and markers of malnutrition such as poor anthropometry measures, as well as low serum albumin, creatinine, hemoglobin, and *n*PCR levels with increased inflammation [191,192,193]. Separately, malnourished HD patients as indicated by MIS ≥ 6 faced 52% increased mortality risk (hazard ratio (HR) = 1.52, 95% CI: 1.13–2.05), along with higher depression symptoms and poorer QoL [194]. These findings suggest that depression should be considered as an independent risk factor for malnutrition. Indeed, malnutrition reversal in HD patients by antidepressant treatment has been observed with significant improvement in *n*PCR, serum albumin, and pre-dialysis blood urea nitrogen levels, along with a significant decrease in depression score compared to healthy controls [206]. Thus, assessment and treatment of depression should be considered as part of overcoming malnutrition in HD patients.

#### 4.5.2. Lack of Social Support

ESKD patients exist in a complex matrix of relationships with family, friends, healthcare professionals, and financial support [207]. The quality of emotional and financial support provided by their social network influences stress management, QoL, health-promoting behaviors, malnutrition, and mortality in HD patients [196,201,207].

HD patients lacking social support have higher prevalence of diminished appetite, reduced physical functioning, and poor adherence to HD treatment [197,200,201]. Greater non-adherence toward nutritional recommendations found in HD patients without family support was due to absence of personal engagement and encouragement from family members to tackle these issues [197,198]. DOPPS Phases 1–3 found that HD patients with poor social support were more likely to experience serum albumin <3.5 g/dL (OR = 1.18, 95% CI: 1.02–1.37) [201]. In contrast, HD patients with social support achieved better social interactions and coping mechanisms toward kidney disease [202], as well as fewer depression symptoms [208]. Ultimately, presence of social support enables patient self-efficacy to reach better health status.

#### 4.5.3. Financial Constraints

Financial constraints commonly faced by HD patients may be attributed to physical limitations to perform work tasks imposed by treatment and time commitments to dialysis treatment [200]. With unemployment, HD patients are dependent on financial support from caregivers or welfare agencies. Incurring financial dependence triggers loss of self-esteem and depression, leading to poor self-efficacy toward health management [195]. The consequence of limited financial resources is suboptimal dietary intake [200].

Employment status in 231 Chinese working-age HD patients was 51% prior to HD initiation, which fell to 11% once they began treatment [209]. These patients reported that the dialysis schedule and post-dialysis fatigue were major reasons for unemployment. HD patients need to strictly adhere to their dialysis schedule of three sessions per week, which consumes up to 18 working h a week [200]. Additionally, fatigue arising post dialysis affects their ability to work [210,211]. Data from the Finnish Registry for Kidney Diseases (*n* = 2637) found that peritoneal dialysis patients compared to HD had a higher employment rate (19% vs. 31%, *p* < 0.001) from greater flexibility in treatment schedule and mobility [212].

Lack of income is commonly prevalent in approximately 50% of the HD population [202,203]. Financial constraints were blamed for poor adherence to dietary recommendations as patients limited their access to healthy food choices on the basis of cost or these patients faced a dilemma on having to choose between spending on medicine or food [202]. Contrarily, HD patients with employment achieved greater DEI (+281 kcal/day, *p* < 0.01) despite perceiving their income to be insufficient to meet their needs [213].

The influence of financial status on dietary intake is unclear but forms a factor contributive to malnutrition [142,213]. Freitas et al. (2014) indicated that low income was associated with a 13% increase in malnutrition prevalence in 344 Brazilian HD patients [203]. Higher SGA and MIS scores of patients indicative of malnutrition were associated with socioeconomic-related nutritional barriers such as difficulty in purchasing food (OR = 1.89, 95% CI: 1.27–2.88, *p* = 0.002) and requiring assistance in meal preparation (OR = 1.15, 95% CI: 1.06–2.06, *p* = 0.001) [166]. Both factors highlighted the impact of financial constraints on nutritional status of HD patients.

#### 4.5.4. Decreased Physical Functioning

Comorbidities associated with CKD such as sarcopenia, vascular dysfunction, inflammation, and malnutrition [214] negatively impacts the three components of physical functioning [215] which are related to body functions and structure, ability to perform, and participation in physical activity.

Fatigue is central to the reduced physical capacity to perform activities of daily living by HD patients and contributes to malnutrition [207,216,217]. The dialysis process itself may cause fatigue, stiffening of joints, and muscle cramping, thus affecting work task performance [218]. Indeed, fatigue was reported to affect the ability to prepare meals as indicated by 59% of HD patients reporting “being too tired to prepare meal” as a barrier toward dietary adherence, and this barrier was associated with lower DEI (*r* = −0.125, *p* = 0.002) [213]. Malnourished HD patients identified using Mini Nutritional Assessment < 19 and SGA > 8 had poor activities of daily living score [219], suggesting that ability to perform simple daily tasks was affected in these patients.

## 5. Conclusions

We presented a global view that malnutrition susceptibility in the HD community is either or both of iatrogenic and of non-iatrogenic origins. Keeping in view disparities in dialysis provision between upper- and lower–middle-income countries, leveraging the point of origin of malnutrition in dialysis patients by healthcare practitioners would enable personalized patient care, as well as country-specific malnutrition treatment strategies. Nutrition assessment is a critical first step to identify the factors of iatrogenic and/or non-iatrogenic origin implicated in malnutrition etiology. This is important as the next step of prevention or treatment for malnutrition depends on the identified factors and aligning effective strategies for nutritional intervention.

The iatrogenic factors that may be implicated in malnutrition may be (i) low dialysis adequacy resulting in poor uremia and metabolic acidosis correction, and/or (ii) low serum albumin levels for patients if dialyzing on highly permeable membranes or dialysis techniques that incur greater amino-acid losses. The non-iatrogenic approach should identify implications of (i) dietary inadequacy as per suboptimal DEIs and DPIs, (ii) poor appetite status, inflammation markers, low diet quality, and high diet monotony index, which indicate barriers to achieving dietary adequacy, and/or (iii) identification of psychosocial and financial barriers to nutritional optimization. These factors are modifiable and should be incorporated as part of a comprehensive nutritional assessment. Identification of factors causing malnutrition that are patient-specific would be crucial for healthcare practitioners to provide a more personalized patient care to treat malnutrition.

## Figures and Tables

**Figure 1 nutrients-12-03147-f001:**
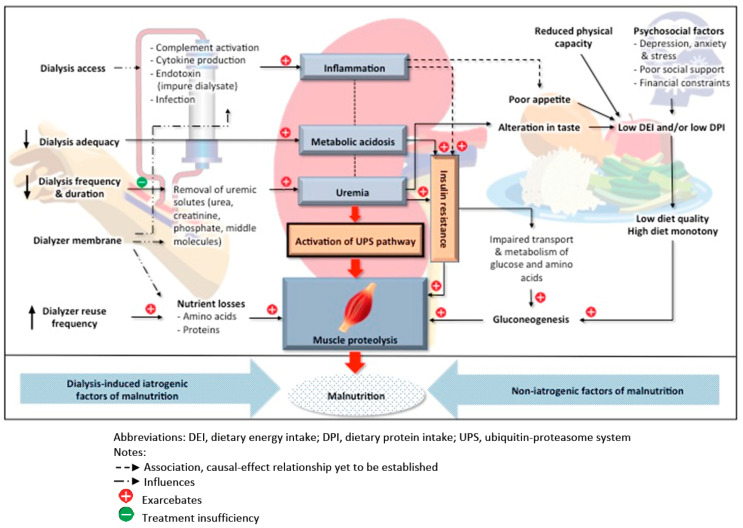
Etiology of malnutrition in dialysis patients.

**Table 1 nutrients-12-03147-t001:** Mortality risk within 12 months of HD initiation according to nutritional indicators of malnutrition.

Reference	Sample Size (*n*)	Predictors of Mortality			
Bradbury et al., 2007 [34]	4802	**Months**	**BMI < 20 kg/m** ^2^	**Ser.Alb < 3.5 g/L**	
		(AHR 95% CI)			
		<4	0.98 (0.67–1.44)	1.57 (1.18–2.09)	
		4–12	1.38 (0.98–1.94)	1.27 (1.00–1.63)	
		>12	1.19 (0.93–1.53)	1.41 (1.17–1.70)	
Lukowsky et al., 2012 [20]	18,707	**Months**	**BMI increase by 2** **index points**	**Ser.Alb < 3.5 g/L**	***n*PCR > 1.0 g/kg/day**
		(AHR 95% CI)			
		<3	0.92 (0.90–0.94)	2.56 (2.30–2.84)	1.21 (1.06–1.38)
		4–6	0.93 (0.91–0.95)	2.04 (1.81–2.31)	0.96 (0.80–1.14)
		7–12	0.94 (0.92–0.96)	1.89 (1.70–2.10)	0.89 (0.74–1.07)
McQuillan et al., 2015 [22]	4807	**Months**		**BMI <18.5 kg/m^2^**	
		<3		AHR (95% CI) = 4.22 (3.12–5.17)	
Murray et al., 2018 [35]	227	**Parameters**	**CPH**	***p*-value**	
		BMI	0.97 (0.85–1.11)	0.625	
		Ser.Alb	0.40 (0.12–1.39)	0.149	
		Undefined malnutrition (using clinical judgment)	4.70 (0.25–88.78)	0.302	

Abbreviations: AHR, adjusted hazard ratio; BMI, body mass index; CI, confidence interval; CPH, Cox proportional hazard; HD, hemodialysis; *n*PCR, normalized protein catabolic rate; Ser.Alb, serum albumin.

**Table 2 nutrients-12-03147-t002:** Mortality risk in maintenance HD patients according to nutritional indicators of malnutrition.

Parameters Associated to Mortality Risk	References
↓ Body mass index	Caetano et al., 2016 [11]; Pifer et al., 2002 [36]
↓ Mid-arm muscle circumference	Araujo et al., 2006 [33]
↓ Fat tissue index	Caetano et al., 2016 [11]
↓ Lean tissue index	Dekker et al., 2016 [37]; Rosenberger et al., 2014 [38]
↓ Serum albumin	Araujo et al., 2006 [33]; Pifer et al., 2002 [36], Caetano et al., 2016 [11]
↓ Serum prealbumin	Chertow et al., 2005 [39]
Modified subjective global assessment (*severe malnutrition*)	Pifer et al., 2002 [36]
↑ Geriatric Nutritional Risk Index	Takahashi et al., 2014 [5]
↓ Dietary energy intake	Araujo et al., 2006 [33]

Abbreviation: HD, hemodialysis. Notes: ↓ decreased trend; ↑ increased trend.

**Table 3 nutrients-12-03147-t003:** Protein and amino-acid losses according to types of dialyzer membranes.

Types of Membrane	Nutrient Losses	References
Cellulosic	7–8 g of amino acids	Wolfson et al., 1982 [42]; Ikizler et al., 1994 [43]
Cellulose acetate with HF	3 g of protein	Honeich et al., 1994 [51]
Cellulose triacetate with HF	4 g of protein	Honeich et al., 1994 [51]
Low flux	5–6 g of amino acids	Ikizler et al., 1994 [43]; Gil et al., 2007 [52]
High flux	5–8 g of amino acids	Ikizler et al., 1994 [43]; Gil et al., 2007 [52]
	3–8 g of protein	Honeich et al., 1994 [51]; Salame et al., 2018 [44]; Ikizler et al., 1994 [43]
Medium cutoff	3–7 g of albumin	Kirsch et al., 2017 [53]
Hemodiafiltration	4–6 g of albumin	Meert et al., 2011 [54]
	9 g of protein	Salame et al., 2018 [44]

Abbreviation: HF, high flux.

**Table 4 nutrients-12-03147-t004:** Effects of type of dialyzer membranes on inflammation status.

Reference	Treatment Duration (Months)	Type of Membrane	Inflammatory Marker Outcomes
Dębska-Ślizień et al., 2014 [80]	6	Polysulfone (low flux)	CRP: 9.3 ± 19.5 to 6.0 ± 6.9 mg/dL
		Polysulfone (high flux)	CRP: 12.2 ± 27.8 to 6.5 ± 9.2 mg/dL
Movili et al., 2015 [81]	12	Usual hemodialysis	CRP: 5.1 ± 6.8 to 5.3 ± 5.0 mg/dL
		Hemodiafiltration	CRP: 6.8 ± 7.0 to 2.3 ± 2.4 mg/dL
Zickler et al., 2017 [66]	1	Polyarylethersulfone/polyvinylpyrrolidone (medium cutoff)	TNF-α: 24.1 ± 8.1 to 20.6 ± 5.8 pg/mL IL-6: 9.0 ± 13.2 to 6.0 ± 5.9 pg/mL CRP: 15.3 ± 30.0 to 9.3 ± 14.5 mg/dL
		Polyarylethersulfone/polyvinylpyrrolidone (high flux)	TNF-α: 23.4 ± 7.3 to 22.0 ± 6.0 pg/mL IL-6: 9.8 ± 20.5 to 5.5 ± 4.5 pg/mL CRP: 13.4 ± 25.5 to 9.6 ± 15.7 mg/dL
Galli et al., 2005 [82]	6	Polymethylmethacrylate (high flux)	TNF-α: 18.7 ± 4.3 to 15.1 ± 3.1 ^a^ pg/mL IL-6: 5.0 ± 1.9 to 3.1 ± 0.6 ^a^ pg/mL CRP: 22.7 ± 33.9 to 12.1 ± 9.1 mg/dL
		Cellulose acetate/cuprammonium rayon (low flux)	TNF-α: 19.0 ± 4.0 to 21.5 ± 5.5 pg/mL IL-6: 5.3 ± 2.1 to 5.8 ± 2.3 pg/mL CRP: 25.8 ± 28.6 to 27.4 ± 24.0 mg/dL

Abbreviations: CRP, C-reactive protein; IL-6, interleukin-6; TNF-α, tumor necrosis factor alpha; ^a^ significantly different (*p* < 0.05) compared to pre-treatment.

**Table 5 nutrients-12-03147-t005:** Inadequate DEI and DPI in global HD populations.

Author/Year	Country	Sample Size, *n*	DEI (kcal/kg BW)/day	DPI (g/kg BW/day)	Dietary Inadequacy *^a^*
*Large Cross-Sectional/Cohort Studies (n > 100)*					
Suaheleen et al., 2020 [142]	Malaysia	382	24.9 ± 5.2	0.90 ± 0.29	DEI: 52% DPI: 40%
Burrowes et al., 2003 [143]	United States	1901	22.70 ± 8.30	0.93 ± 0.35	-
Harvinder et al., 2013 *^b^* [144]	Malaysia	155	25.5 ± 8.5	1.07 ± 0.47	DEI: 75% DPI: 67%
Ichikawa et al., 2007 *^b^* [145]	Japan	200	29.3	1.08 ± 0.17	-
Kang et al., 2017 [138]	Korea	144	25.8 ± 5.4	0.88 ± 0.23	-
Moreira et al., 2013 [146]	Portugal	130	25.8	1.27	DEI: 74.6% DPI: 32.3%
Rocco et al., 2002 [147]	United States	1000	22.90 ± 8.40	0.93 ± 0.36	DEI: 92% DPI: 81%
Sahathevan et al., 2015 [148]	Malaysia	205	23.12 ± 6.94	0.94 ± 0.39	DEI: 65% DPI: 42%
*Small-Scale Studies (n < 100)*					
Adanan et al., 2019 [149]	Malaysia	54	21.8 ± 4.8	0.7 ± 0.2	-
Arslan and Kiziltan, 2010 [150]	Turkey	93	34.20 ± 8.89	0.94 ± 0.26	-
Chauveau et al., 2007 [151]	France	99	29.80 ± 7.50	1.18 ± 0.28	-
Johansson et al., 2013 *^b^* [152]	England	53	24.30 ± 6.70	0.97 ± 0.25	-
Kalantar-Zadeh et al., 2002 [153]	United States	30	26.40 ± 15.30	0.88 ± 0.57	-
Kim et al., 2015 [154]	Korea	63	21.90 ± 6.70	0.90 ± 0.30	-
Morais et al., 2005 [155]	Brazil	44	20.70 ± 6.70	1.20 ± 0.60	-
Shapiro et al., 2015 [156]	United States	13	25.4 ± 7.4	1.03 ± 0.32	-
Vijayan et al., 2014 [157]	India	98	31.3	0.98	-

Abbreviations: BW, body weight; DEI, dietary energy intake; DPI, dietary protein intake; HD, hemodialysis. *^a^* Cutoff for dietary inadequacy: DEI < 35 kcal/kg BW/day; DPI < 0.8 g/kg BW/day. *^b^* Used ideal body weight for calculation of dietary adequacy.

**Table 6 nutrients-12-03147-t006:** Suboptimal dietary intakes and mortality in HD patients.

Author/Year	Country	Patient No.	Follow-Up	DEI (kcal/ kg BW)/day		Hazard Ratio (95% CI)	DPI (g/kg BW/day)		Hazard Ratio (95% CI)
				Survivors	Non-Survivors		Survivors	Non-Survivors	
Antunes et al., 2010 *^a^* [159]	Brazil	79	33 (17–38) months	25.9 (22.0–29.8)	22.0 (18.0–26.0) *^b^*	-	1.20 (0.86–1.47)	0.93 (0.90–1.1)	DPI < 1.2g/kg: 4.98 (1.47–16.86) *^b^*
Araujo et al., 2006 [33]	Brazil	344	10 years	27.4 ± 8.9	23.5 ± 7.4 *^b^*	0.96 (0.92–0.99) *^b^*	1.01 ± 0.38	0.92 ± 0.34 *^c^*	-
Beberashvili et al., 2011 [164]	Israel	85	2 years	20.8 ± 5.4	19.1 ± 1.4	-	0.88 ± 0.24	0.81 ± 0.10	-
Kang et al., 2017 [138]	Korea	144	10 years	26.7 ± 5.8	24.3 ± 4.2 *^b^*	DEI <25 kcal/kg: 1.86 (1.02–3.40) *^b^*	0.91 ± 0.21	0.82 ± 0.24 *^b^*	DPI < 0.8g/kg: 1.35 (0.77–2.35)

Abbreviations: BW, body weight; CI, confidence interval; DEI, dietary energy intake; DPI, dietary protein intake; HD, hemodialysis. *^a^* Inclusive of HD and peritoneal dialysis patients. *^b^* Significant at *p* < 0.05. *^c^* Significant at *p* < 0.01.

**Table 7 nutrients-12-03147-t007:** Nutritional outcomes associated with anorexia and appetite hormones in HD patients.

Associations	References
**Poor Appetite**	
↓ BMI, MAC, MAMC, and MAMA	Bossola et al., 2011 [167]; Sahathevan et al., 2015 [148]
↓ Muscle mass as per BCM, LTM, and LBM index measures	Ekramzadeh et al., 2014 [166]; Sahathevan et al., 2015 [148]; Oliveira et al., 2015 [100]
↓ Serum albumin, serum prealbumin and *n*PCR/*n*PNA	Molfino et al., 2015 [177]; Oliveira et al., 2015 [100]; Bossola et al., 2011 [167]; Kalantar-Zadeh et al., 2004 [163]
↑ hsCRP	Sahathevan et al., 2015 [148]; Kalantar-Zadeh et al., 2004 [163]
↑ DMS, MIS and PG-SGA	Sahathevan et al., 2015 [148]
	Ekramzadeh et al., 2014 [66]; Kalantar-Zadeh et al., 2004 [163]
↓ GNRI	Oliveira et al., 2015 [100]
↓ Overall food intake of < 50%	Molfino et al., 2015 [177]
↓ DEI and DPI	Sahathevan et al., 2015 [148]
↑ **Ghrelin**	
↓ BMI	Mafra et al., 2010 [175]
↓ Serum albumin and *n*PNA	Perez-Fontan et al., 2004 [178]
↑ MIS	Vanita et al., 2016 [170]
↓ SGA	Perez-Fontan et al., 2004 [178]
↑↓ **Leptin**	
↑ BMI, ↑ leptin	Montazerifar et al., 2015 [173]
↑ Serum albumin, ↑ leptin	Montazerifar et al., 2015 [173]
↑ DMS and MIS, ↓ leptin	Kursat et al., 2010 [179]; Ko et al., 2020 [172]

Abbreviations: BCM, body cell mass; BMI, body mass index; DEI, dietary energy intake; DMS, Dialysis Malnutrition Score; DPI, dietary protein intake; GNRI, Geriatric Nutritional Risk Index; HD, hemodialysis; hsCRP, high-sensitivity C-reactive protein; LBM, lean body mass; LTM, lean tissue mass; MAMA, mid-arm muscle area; MAC, mid-arm circumference; MAMC, mid-arm muscle circumference; MIS, malnutrition–inflammation score; *n*PCR, normalized protein catabolic rate; *n*PNA, normalized protein nitrogen appearance, PG-SGA, patient-generated subjective global assessment; SGA, subjective global assessment. Notes: ↓ decreased trend; ↑ increased trend.

**Table 8 nutrients-12-03147-t008:** Impact of psychosocial factors on nutritional outcomes.

Associations	References
**Depression**	
↓ BMI, TSF, and MAMC	Koo et al., 2003 [191]
↓ Serum albumin, creatinine, hemoglobin, *n*PCR	Koo et al., 2003 [191]; Choi et al., 2012 [192]; Ogrizovic et al., 2008 [193]
↑ Inflammation markers	Choi et al., 2012 [192]; Ogrizovic et al., 2008 [193]
↓ SGA	Koo et al., 2003 [191]
↑ MIS	Lopes et al., 2017 [194]
↓ QoL	Natashia et al., 2019 [195]
**Lack of social support**	
↓ Adherence to dietary restriction	Kiajamali et al., 2017 [196]; Kara et al., 2007 [197]; Dilek and Kocaoz, 2015 [198]; Aness et al., 2018 [199]
↓ Appetite	Lopes et al., 2007 [200]
↓ QoL	Untas et al., 2011 [201]
**Financial constraints**	
↓ Access to purchase food	Clark-Cutaia et al., 2018 [202]; Ekramzadeh et al., 2014 [166]
↓ Adherence to dietary restriction	Clark-Cutaia et al., 2018 [202]
↑ SGA	Freitas et al., 2014 [203]
↑ MIS	Freitas et al., 2014 [203]

Abbreviations: BMI, body mass index; MAC, mid-arm circumference; MAMC, mid-arm muscle circumference; MIS, malnutrition–inflammation score; *n*PCR, normalized protein catabolic rate; QoL, qualify of life; SGA, subjective global assessment; TSF, triceps skinfold. Notes: ↓ decreased trend; ↑ increased trend.

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
