# Peer review of "Understanding Development of Malnutrition in Hemodialysis Patients: A Narrative Review"

_nutrients, 2020, doi:10.3390/nu12103147_

Round 1
Reviewer 1 Report
The authors review the well-known nutrition problems and guidelines in dialysis patients.
Remarks:
Table 3: in the article of NDT 2017 Kirsch, the protein loss was notably higher in medium cut-off compared to HDF. In HDF, a loss of 14g/dialysis session of 4hours doesn’t have to occur or the dialyzer should not be used in HDF but in HD.
Page 10: “dialysis adequacy [103,104] through maintaining the pre-dialysis serum bicarbonate levels between 24-26 mmol/L [107 “. The threshold for pre-dialysis bicarbonate is set too high and should not exceed 24 mmol/L. The risk is to reach a symptomatic alkalosis level at the end of the dialysis session that ensues vomiting and hypotension. Madhukar M Nephrol Dial Transplant 2016; 31: 1220-1224, Bommer JLF, Am J Kidney Dis 2004; 44: 661-671; Tentori FKA Am J Kidney Dis 2013; 62: 738-746, Wu DSC, Clin J Am Soc Nephrol 2006; 1: 70-78; Vashisha T Clin J Am Soc Nephrol 2013; 8:254-264; Basile C, Kidney Int 2016; 89: 1008-1015
Page 11 : “In one study, patients with serum bicarbonate levels ≤22 mmol/L rather than serum bicarbonate levels >22 mmol/L had lower serum albumin levels (p=0.046)”. Usually the higher is the protein intake, the lower is the serum bicarbonate because as in non-dialysis patients, the protein intake increases the acid load. Therefore the acidosis in dialysis patients is often associated with a better nutritional status. After adjustment for the nutritional status, acidosis is often associated with an increased risk of mortality as bicarbonate > 24 mmol/L. Correction of the metabolic acidosis by increasing the bicarbonate concentration of the dialysate seems not adequate and is also associated with an increased mortality risk. See the review in NDT of major dialysis cohorts. But the level of evidence based medicine in this topic is scarce. Any randomized study can fully answer to the question of the ideal bicarbonate level in dialysis patients but a level of 22-24 mmol/l has been recommended.
Page 11: “Contrarily, serum bicarbonate levels >27 mmol/L promote mortality risk in malnourished HD patients [“ Even a level higher than 24 mmol/L has been associated with an increased risk of mortality.
Page 17: “Therefore, insulin resistance promotes muscle proteolysis and this association is evident in HD patients with studies reporting an inverse relationship between insulin resistance and muscle loss [167,168].” When the insulin resistance increases, the muscle loss increases. So it’s not an inverse correlation but a positive correlation.
Reference 48: “Florens, N.; Julliard, L. Large Middle Molecule and Albumin Removal: Why Should We Not R Vanholder, R.; Pletinck, A.; Schepers, E.; Glorieux, G. Biochemical and Clinical Impact of Organic Uremic Retention Solutes: A Comprehensive Update. Toxins. 2018, 10(1), 33.
It seems that you mixed two different references:
Toxins (Basel). 2018 Jan; 10(1): 33. Biochemical and Clinical Impact of Organic Uremic Retention Solutes: A Comprehensive Update Raymond Vanholder,* Anneleen Pletinck, Eva Schepers, and Griet Glorieux
And Contrib Nephrol 2017;191:178-187. Large Middle Molecule and Albumin Removal: Why Should We Not Rest on Our Laurels? Nans Florens, L Juillard
Reference :” Chazot, C.; Jean, G. The advantages and challenges of increasing the duration and frequency of 688 maintenance dialysis sessions. Nature. 2009,5, 34-44. 689” Nature clin pract nephrol and not nature

Author Response
| Reviewer comment | Author's feedback |
|
Table 3: in the article of NDT 2017 Kirsch, the protein loss was notably higher in medium cut-off compared to HDF. In HDF, a loss of 14g/dialysis session of 4hours doesn’t have to occur or the dialyzer should not be used in HDF but in HD. |
We appreciate and agree with this comment. The type of membrane used with dialysis technique does make a difference as the combination of dialyzer membrane with greater seiving coefficient with HDF technique has been shown to introduce greater albumin losses as indicated by Krieter et al. (2010). As previously mentioned in Lines 163-166, the combination use of highly permeable membrane (MCO or HCO) with HF or HDF technique is not encouraged. We also acknowledge that the extent of albumin loss as per Kirsh et al. (2017) was wrongly stated in Table 3. Corrections have been made accordingly to Table 3. |
|
Page 10:
“dialysis adequacy [103,104] through maintaining the pre-dialysis serum bicarbonate levels between 24-26 mmol/L [107 “. The threshold for pre-dialysis bicarbonate is set too high and should not exceed 24 mmol/L. The risk is to reach a symptomatic alkalosis level at the end of the dialysis session that ensues vomiting and hypotension. Madhukar M Nephrol Dial Transplant 2016; 31: 1220-1224, Bommer JLF, Am J Kidney Dis 2004; 44: 661-671; Tentori FKA Am J Kidney Dis 2013; 62: 738-746, Wu DSC, Clin J Am Soc Nephrol 2006; 1: 70-78; Vashisha T Clin J Am Soc Nephrol 2013; 8:254-264; Basile C, Kidney Int 2016; 89: 1008-1015 |
The threshold of 24-26 mmol/L as stated in the manuscript refers to the current opinion of NKF-KDOQI Clinical Practice Guideline for Nutrition in CKD: 2020 Update. The literature you have cited is based from 2016 and earlier providing the pre-dialysis serum bicarbonate at more than 22 mmol/L, which was proposed by the earlier KDOQI Guideline (2000). We prefer to use the current opinion by the NKF-KDOQI Guidelines. In the text, we have amended the sentence to: Section 3.8: Efficacy of Metabolic Acidosis Correction, Line 308-309: HD treatment aims to correct metabolic acidosis via bicarbonate concentration of the dialysate [124], ultrafiltration rate [125], dialyzer membrane surface area and permeability [126], blood and dialysis flow rate [125], trans-membrane concentration gradient set by the patient’s serum bicarbonate level and bicarbonate availability from the dialysate [126], as well as dialysis adequacy [123,124] through maintaining the pre-dialysis serum bicarbonate levels between 24-26 mmol/L as recommended by current opinion [127]. |
|
Page 11 : “In one study, patients with serum bicarbonate levels ≤22 mmol/L rather than serum bicarbonate levels >22 mmol/L had lower serum albumin levels (p=0.046)”. Usually the higher is the protein intake, the lower is the serum bicarbonate because as in non-dialysis patients, the protein intake increases the acid load. Therefore the acidosis in dialysis patients is often associated with a better nutritional status. After adjustment for the nutritional status, acidosis is often associated with an increased risk of mortality as bicarbonate > 24 mmol/L. Correction of the metabolic acidosis by increasing the bicarbonate concentration of the dialysate seems not adequate and is also associated with an increased mortality risk. See the review in NDT of major dialysis cohorts. But the level of evidence based medicine in this topic is scarce. Any randomized study can fully answer to the question of the ideal bicarbonate level in dialysis patients but a level of 22-24 mmol/l has been recommended. |
We conquer with the reviewer’s comment and given the pausity of studies in this area, there is a grey area in terms of the ideal bicarbonate level. In acknowledgement, we have added a qualifying statement in the text as follows (Lines 338-342): In one study, patients with serum bicarbonate levels ≤22 mmol/L rather than serum bicarbonate levels >22 mmol/L had lower serum albumin levels (p=0.046) [135]. In these patients, high serum bicarbonate levels negatively correlated with nPCR (r=-0.492, p=0.045) but positively with serum albumin (r=0.432, p=0.019). Acidosis-led catabolism triggers breakdown of the endogenous proteins, which influence higher nPCR levels. In different malnourished HD populations, a serum bicarbonate level of >23 [136,137] or >27 mmol/L [128,137] have been associated with greater mortality risk. As malnutrition is a confounding factor for serum bicarbonate level, there is no an ideal serum bicarbonate level that fits for all dialysis patients [138].
|
|
Page 11: “Contrarily, serum bicarbonate levels >27 mmol/L promote mortality risk in malnourished HD patients [“ Even a level higher than 24 mmol/L has been associated with an increased risk of mortality. |
We take note of this comment and the associated literature and have amended our text as follows (Lines 338-340): In different malnourished HD populations, a serum bicarbonate level of >23 [136,137] or >27 mmol/L [128,137] have been associated with greater mortality risk.
137. Bommer, J.; Locatelli, F.; Satayathum, S.; Keen, M.L.; Goodkin, D.A.; Saito, A.; Akiba, T.; Port, F.K.; Young, E.W. Association of Predialysis Serum Bicarbonate Levels With Risk of Mortality and Hospitalization in the Dialysis Outcomes and Practice Patterns Study (DOPPS). Am J Kidney Dis. 2004,44(4):661-71. |
|
Page 17: “Therefore, insulin resistance promotes muscle proteolysis and this association is evident in HD patients with studies reporting an inverse relationship between insulin resistance and muscle loss [167,168].” When the insulin resistance increases, the muscle loss increases. So it’s not an inverse correlation but a positive correlation. |
We have taken note of this comment and amended Line 462 to: Therefore, insulin resistance promotes muscle proteolysis and this association is evident in HD patients with studies reporting a positive correlation between insulin resistance and muscle loss [167,168]. |
|
Reference 48: “Florens, N.; Julliard, L. Large Middle Molecule and Albumin Removal: Why Should We Not R Vanholder, R.; Pletinck, A.; Schepers, E.; Glorieux, G. Biochemical and Clinical Impact of Organic Uremic Retention Solutes: A Comprehensive Update. Toxins. 2018, 10(1), 33. It seems that you mixed two different references: Toxins (Basel). 2018 Jan; 10(1): 33. Biochemical and Clinical Impact of Organic Uremic Retention Solutes: A Comprehensive Update Raymond Vanholder,* Anneleen Pletinck, Eva Schepers, and Griet Glorieux And Contrib Nephrol 2017;191:178-187. Large Middle Molecule and Albumin Removal: Why Should We Not Rest on Our Laurels? Nans Florens, L Juillard |
We take note of this comment and have amended the reference [57] correctly to: Florens, N.; Juillard, L. Large Middle Molecule and Albumin Removal: Why Should We Not Rest on Our Laurels?Contrib Nephrol. 2017, 191,178-187. Vanholder et a. (2018) remains as reference 107. |
| Reference :” Chazot, C.; Jean, G. The advantages and challenges of increasing the duration and frequency of 688 maintenance dialysis sessions. Nature. 2009,5, 34-44. 689” Nature clin pract nephrol and not nature |
We take note of this comment and have amended the reference [116] to: Chazot, C.; Jean, G. The advantages and challenges of increasing the duration and frequency of maintenance dialysis sessions. Nature Clin Pract Nephrol. 2009,5, 34-44. |

Reviewer 2 Report
The authors extensively review the current evidence of iatrogenic and non-iatrogenic causes of malnutrition in hemodialysis patients. The manuscript is well written and informative and will be of great interest to readers of the journal. Only some reference errors have to be revised (In Table 1, cited Reference 25 should be 24, and reference 26 should be 25). In addition, in page 9, line 108, "total renal clearance" should be revised to "total clearance".
Author Response
| Reviewer comment | Author's feedback |
| The authors extensively review the current evidence of iatrogenic and non-iatrogenic causes of malnutrition in hemodialysis patients. The manuscript is well written and informative and will be of great interest to readers of the journal. | Thank you for your feedback. |
| Only some reference errors have to be revised (In Table 1, cited Reference 25 should be 24, and reference 26 should be 25). | We take note and corrections have been made to Table 1 accordingly. |
| In addition, in page 9, line 108, "total renal clearance" should be revised to "total clearance". |
We have amended “total renal clearance" in Line 247 to "total clearance”. |

Reviewer 3 Report
Very interesting review about malnutrition in HD patients.
In paragraph 2 (start of dialysis) please comment on the use of nutritional factors like nPNA in the decision to start dialysis.
Please expand on the interplay of malnutrition and oxidative stress in HD
Please include the Malnutrition Inflammation Atherosclerosis syndrome in the review.
Please comment on the role of dialysis access on malnutrition
Please comment on the role of heart failure on malnutrition. Similarly on the role of comorbidities on malnutrition
Some linguistic editing is necessary.
Author Response
| Reviewer's comment | Author's comment |
| In paragraph 2 (start of dialysis) please comment on the use of nutritional factors like nPNA in the decision to start dialysis. |
The following comment was added to Section 2. Development of Malnutrition at the Time of HD Initiation and Indicators of Poor Nutritional Status (Lines 97-107):
Earlier opinion on dialysis initiation did consider poor nutritional status as a factor to initiate dialysis. However, this was not based on markers of malnutrition but rather signs and symptoms of malnutrition such as anorexia, nausea and fatigue [24]. Van de Luijtgaarden et al. (2012) reported 53% of nephrologists agreeing to initiate dialysis in patients with poor nutritional status [25]. However, a review on dialysis initiation observed the lack of data on benefits of early dialysis initiation in patients with low serum albumin level or in improving nutritional status [26]. The Canadian Society of Nephrology Guidelines (2014) [27] ceased to recommend dialysis initiation based on decline in nutritional status as indicated by serum albumin, lean body mass or Subjective Global Assessment), whereas the Caring for Australians with Renal Impairement Guidelines [28] do recommend dialysis with GFR <10 mL/ min per 1.73 m2 to reduce uremic symptoms or signs of malnutrition.
|
| Please expand on the interplay of malnutrition and oxidative stress in HD |
The following paragraphs have been added to detail the interplay of malnutrition with oxidative stress (Section 3.3 Dialysis-induced Inflammation: Lines 211-225) Another impact of the HD treatment is the activation of polymorphonuclear white blood cells, which trigger production of ROS and other pro-oxidants [91]. Indeed, increased indices of oxidative damage along with decreased indices of antioxidant defense have been observed in HD patients post-dialysis [92]. Low antioxidant levels in HD patients may also occur from limited vegetable and fruit intakes to prevent hyperkalemia [93]. Resultant low intakes of vitamin A, C, E, and selenium would affect antioxidant defence mechanisms [93.94]. Additionally, involuntary removal of vitamins also occurs with every HD session as mentioned in Section 3.1. However, Silva et al. (2019) found no different in markers of oxidative stress and antioxidant defences in malnourished HD patients identified using Global Objective Assessment compared to mild or well-nourished patients [95]. In contrast, accumulation of advanced glycation end products, a biomarker of oxidative stress measured using skin auto fluorescence was significantly associated with markers of malnutrition in HD patients such as lower serum albumin, lower handgrip strength and lower protein intake [96]. HD patients with protein-energy wasting were 5.2 times more likely to experience oxidative stress as demonstrated by high protein carbonyl levels (95% CI: 24.0-1.1,p=0.039) [97].
|
| Please include the Malnutrition Inflammation Atherosclerosis syndrome in the review. |
We do not wish to discuss MIA syndrome in this review as: MIA syndrome only occurs with the association between inflammation, malnutrition and atherosclerosis. Our review focuses on the development of malnutrition and the categorization of contributing causes into iatrogenic and non-iatrogenic factors. This classification is overarching to include inflammation [Section 3.3 Dialysis-induced inflammation] is the iatrogenic contributive factor of malnutrition but cannot include atherosclerosis. For this reason, our review area has been explained in Lines 57-61. |
| Please comment on the role of dialysis access on malnutrition |
We have added the following paragraphs to Section 3.3 Dialysis-Induced Inflammation, Lines 180-190: Whether dialysis access directly contributes to malnutrition has not been shown. Arteriovenous fistula (AVF) failure were not influenced by markers of poor nutritional status except for high cholesterol (p=0.034) and low nPCR levels (p=0.029) [77]. HD patients with catheter access compared to fistula and graft had significantly higher MIS scores and lower serum albumin levels [78]. In fact, AVF have 52% greater survival rate compared to those on central venous catheter (CVC) irrespective of nutritonal status although malnutrition was found to lower survival rate by 2% [79]. Instead, catheter rather graft and fistula access appears to be a significant predictor of greater inflammatory response and associated with the highest all cause-mortality rate [78] mediated by infection [20]. Therefore, the route of dialysis access is rather associated with inflammation and mortality risk [78], whereby presence of malnutrition may influence the survival rate [79].
|
| Please comment on the role of heart failure on malnutrition. Similarly on the role of comorbidities on malnutrition |
We have added this paragraph to Introduction, Lines 63-70:
Comorbidities such as heart failure (left ventricular failure) and CKD-mineral bone disorder have a bidirectional association with nutritional status [13,14]. The related mechanisms includes malabsorption due to gut edema, poor appetite due to cytokine production, difficulty in oral intake and food preparation arising from fatigue and breathing difficulty [15]. However, malnutrition is acknowledged to be a kidney-specific risk factor for heart failure in CKD patients [14], and with HD patients there is lack of association between malnutrition assessment and echocardiographic findings [16] or CVD risk [17].
heart failure patients. J Cardiovasc Nurs. 2008, 23(4), 357-363.
|
| Some linguistic editing is necessary | We take note of this request and this manuscript was submitted for expert English language editing. |

Round 2
Reviewer 1 Report
no other comments